# Cytomegalovirus Pneumonia in a Patient with Down Syndrome

**DOI:** 10.3390/medicina60020242

**Published:** 2024-01-30

**Authors:** Francesco Perrotta, Donato Piscopiello, Daniela Rizzo, Gaetano Iosa, Giorgio Garzya, Piero Calò, Daniele Gemma

**Affiliations:** Department of Anesthesia and Intensive Care, Azienda Ospedaliera Cardinale Panico, 73039 Tricase, Italy; dpiscopiello@piafondazionepanico.it (D.P.); d.rizzo@piafondazionepanico.it (D.R.); g.iosa@piafondazionepanico.it (G.I.); g.garzya@piafondazionepanico.it (G.G.); p.calo@piafondazionepanico.it (P.C.); d.gemma@piafondazionepanico.it (D.G.)

**Keywords:** Down syndrome, pneumonia, cytomegalovirus

## Abstract

Down syndrome (DS) is a chromosomal disorder due to the presence of an additional chromosome 21 that causes intellectual deficit and physical anomalies and predisposes patients to develop infections throughout their lives. Pneumonias are more serious in patients with DS, requiring hospitalization, and they represent an important cause of mortality in this population. Cytomegalovirus (CMV) causes widespread and serious infections in immunocompromised individuals, affecting the respiratory tract and, when causing interstitial pneumonia, associated with a high mortality rate. However, CMV-induced pneumonia is not reported in DS patients. The prevalence and severity of CMV respiratory infections in subjects with DS is unknown. This case describes a 50-year-old female patient with DS who developed extensive bilateral pneumonia with severe respiratory failure which required hospitalization in intensive care, intubation, and mechanical ventilation after approximately 10 days of empiric antibiotic and anitimycotic therapy for fever, cough, and dyspnea. The patient was diagnosed with CMV pneumonia and recovered after treatment with ganciclovir. To the best of our knowledge, this is the first reported case of CMV pneumonia in a patient with DS. This case aims to highlight that CMV pneumonia in individuals with DS can be a life-threatening condition. It also clarifies the importance of early diagnosis of infections from opportunistic pathogens such as CMV to ensure timely and efficient treatment.

## 1. Introduction

Down syndrome (DS) represents the most frequent chromosomal disease. This genetic disorder is linked to the presence of an extra copy of chromosome 21, and the current prevalence is estimated to be 1.44 per 1000 live births in the United States [1]. There are several comorbidities that contribute to a shorter life expectancy in those with DS compared to the healthy population; however, respiratory failure, pneumonia, and heart failure are the risk factors identified for in-hospital mortality.

In an extensive review of medical records of patients with DS between 2011 and 2020, the age groups most frequently admitted to hospital were those under 5 years and those over 45 years. The majority of these patients required supplemental oxygen, with 46% requiring admission to an intensive care unit [2].

Susceptibility to pneumonia and other infections is probably multifactorial in nature and could be due to both immune system dysfunction and non-immunological factors.

In adult patients, the immune disorders underlying increased susceptibility to respiratory infections are poorly understood [3].

Similarly, there are few studies describing the pathogens most involved in the etiology of pneumonia in patients with DS.

In a 2020 study, Santoro pointed out that in a review of articles on pneumonia in patients with DS, the infectious etiology was provided in only one article and did not show a consistent pathogen but, rather, a variety of causes [4]. The identification of the specific infectious etiology could have important implications for prevention, the choice of therapy, and identification of the underlying risk factor.

Cytomegalovirus (CMV) is a DNA virus member of the Herpesviridae family, and is considered to be an opportunistic pathogen responsible for life-threatening community-acquired pneumonia in immunosuppressed patients, for example, transplant recipients and those taking immunosuppressive drugs or steroids. Similarly, in patients with DS, cytomegalovirus pneumonia could be a complication of CMV disease favored by immune disorders.

CMV pneumonia is characterized by the presence of nonspecific clinical signs and symptoms combined with the detection of CMV from the lung [5]. Patients with CMV pneumonia typically present with nonproductive cough, dyspnea, fever, and hypoxia, none of which are typically associated with CMV pneumonia but are only indicative of an ongoing pulmonary process, and this can lead to a dangerous diagnostic delay in the categories of patients at risk.

The diagnostic tests for detecting the virus are the same as those used to diagnose CMV infection in transplant patients. The standard diagnostic method involves detection and quantification of the CMV genome using quantitative nucleic acid amplification tests (QNATs), which are very sensitive and allow for rapid turnaround times. Cutoff values have been indicated for bronchoalveolar lavage (BAL) specimens that differentiate between pneumonia and lung shedding in hematopoietic stem cell transplant (HSCT) patients, but these values are likely variable depending on the test used and the population undergoing tests [6]. Human cytomegalovirus (HCMV) blood or plasma QNAT is also useful to support the diagnosis of probable CMV pulmonary or gastrointestinal disease, or when it is risky to perform a lung biopsy [6,7].

We describe a clinical case of a 50-year-old patient with Down syndrome and renal failure who arrived at our intensive care unit with severe ARDS, after having been treated initially at home with empiric antibiotic therapy and subsequently in the medical department of another hospital with antibiotics and antifungals for bilateral pneumonia.

The patient, with an unknown history of colonization or previous cytomegalovirus infections, was diagnosed with CMV pneumonia using molecular testing on BAL fluid and recovered after treatment with ganciclovir.

We obtained the family’s consent for publication and the approval of our institution.

## 2. Case Presentation

A 50-year-old patient with Down syndrome came to our ED with fever (38 °C), severe respiratory failure, tachypnea and dyspnea, an SpO2 of 82% with Venturi mask valves 15l/FiO2 60%, sinus tachycardia, blood pressure of 105/63 mmHg, and imaging of bilateral pneumonia in an X-ray of the chest, with greater involvement of the right lung (Figure 1). The patient came from the medical department of another hospital, and the symptoms had started about 10 days prior with a cough and low-grade fever. After seven days of home therapy with ceftriaxone and azithromycin for worsening symptoms, she was admitted to hospital with the diagnosis of bilateral interstitial pneumonia. Here, empiric therapy with meropenem, linezolid, and caspofungin was started, and supplemental oxygen was required.

After approximately 36 h, due to the worsening of her clinical condition, she was transferred to our emergency room where she arrived in the clinical condition previously described. Her medical history described renal failure, there was no history of a previous cytomegalovirus infection, and she was in good hemodynamic compensation.

Severe hypoxia despite supplemental oxygen and respiratory distress required orotracheal intubation and mechanical ventilation in the emergency room. A chest CT scan was then performed, which showed extensive consolidation of the right lung and subtotal consolidation of the left lung (Figure 2).

Upon admission to the intensive care unit, blood tests showed leukocytosis (16,060 cells/μL), C-reactive protein (22.32 mg/dL), procalcitonin (3.22 ng/mL), and the presence of anemia (Hb 10 g/dL) with vitamin B12 deficiency (81 pg/mL) and renal failure (creatinine 1.63 mg/dL).

A urine sample was analyzed for urinary antigens for legionella and streptococcus, with negative results.

The antibiotic and antifungal therapy with meropenem, linezolid, and caspofungin, which had begun approximately 36 h earlier at another hospital, was continued.

The patient underwent transesophageal echocardiography, and Pro-BNP was measured to exclude congestive heart failure.

Fibro-bronchoscopy was performed, which highlighted a hyperemic tracheo-bronchial mucosa and absence of tracheo-bronchial secretions; at the same time, a broncholavage sample was sent to bacteriology for molecular diagnostics with a Filmarray of the lower respiratory tract, for phenotypic culture examination and for galactomannan testing.

No microorganisms were isolated on blood cultures or BAL fluid. The PCR tests were negative for galactomannan and for the presence of Pneumocystis jirovecii, Mycoplasma pneumoniae, herpes simplex and syncytial viruses, and Mycobacterium tuberculosis. COVID-19 pneumonia was excluded with a negative molecular test for SARS-CoV-2.

In consideration of the clinical history and the poor response to the ongoing therapy, with persistent hypoxia, a second BAL was performed 48 h after the first to search for cytomegalovirus DNA (CMV DNA), which gave a positive result but with a low load (300 copies/mL) of 2.4 log CMV DNA.

Ganciclovir therapy was therefore started at the correct dosage for renal function indices.

After 48 h of therapy with ganciclovir, a progressive and rapid reduction in white blood cells (WBC), C-reactive protein, creatinine, and procalcitonin was observed. (Table 1).

Likewise, an improvement in respiratory exchange and radiological images was observed (Figure 3).

Mechanical ventilation was stopped after 10 days (Figure 4), and the patient was transferred to the medical department where she continued therapy with ganciclovir for another two weeks before being discharged to her home.

## 3. Discussion

Subjects with DS have a higher incidence of pneumonia than do people without DS, often requiring hospitalization and intensive care. Frequent comorbidities associated with DS, such as heart disease and neurological disease in adulthood, only partially influence the effect of DS on pneumonia [8].

In this study, we described the case of an adult patient with DS with a severe form of bilateral pneumonia and BAL fluid positive for CMV DNA.

Cytomegalovirus pneumonia is a rare but life-threatening complication of CMV disease. Several clinical conditions put patients at high risk for HCMV infection, which may favor the onset of interstitial pneumonia. Pneumonia is the most common HCMV infection in hematopoietic stem cell transplant (HSCT) patients, with high mortality rates [9,10]. Likewise, solid organ transplant patients are at high risk of developing HCMV lung infection [11,12]. In patients who have received lung transplantation, HCMV pneumonia can occur despite prophylaxis and is associated with adverse outcomes [13]. CMV interstitial pneumonia is also frequent among HIV-infected patients [14] and may be the first manifestation of severe combined immunodeficiency (SCID) [15].

The aforementioned high-risk clinical conditions for the development of CMV pneumonia are all associated with reduced T cell immunity, indicating a significant role for this type of immune cell.

However, infrequent cases of CMV pneumonia have also occurred in non-immunosuppressed patients, suggesting that pathogenicity determinants encoded by the virus may also cause lung disease [16,17,18].

CMV interstitial pneumonia causes nonspecific symptoms such as dry cough, dyspnea, exertional dyspnea, and fever [18]. Radiological images are also rather nonspecific and include bilateral interstitial infiltrates on chest radiography and ground-glass opacities, with small nodules, on a chest CT scan [19]. Clinical and radiological images are therefore nonspecific and are also common to other forms of interstitial pneumonia, which can lead to delays in the diagnosis of cytomegalovirus pneumonia [20].

The diagnosis of CMV pneumonia must therefore be supported by the detection of virus DNA in blood and/or bronchoalveolar lavage samples. In our case, on the basis of clinical suspicion and the lack of response to antibiotic and antifungal therapy, we performed the search for CMV DNA only on the BAL fluid, without performing the search on blood, and we did not perform serological tests.

Caution should be exercised when interpreting diagnostic test results for HCMV.

Doctors should evaluate the results together with radiological images, medical history, and symptoms and signs. It is not always easy to distinguish between an active, reactivated, or latent CMV infection. A healthy person with a previous CMV infection will continue to harbor the virus throughout their life with the possibility of intermittent reactivation and with small quantities of the virus being found in body fluids without causing symptoms.

Furthermore, immunosuppressed individuals may not have a strong antibody response to CMV infection, so IgM and IgG levels may be lower than expected even if they have active CMV disease.

Likewise, CMV may not be present in sufficient quantities in a particular fluid or tissue biopsy to be detected.

Tests performed in the very early stages of an infection may not detect the presence of CMV antibodies.

A definitive diagnosis of CMV pneumonia requires a lung biopsy demonstrating the cytopathic effect of the virus with typical intranuclear and intracytoplasmic inclusions and allowing CMV-specific immunostaining [5].

However, lung biopsy is often not possible due to the high risk of bleeding and pneumothorax [21].

Detection of CMV DNA in blood or plasma, although less invasive, has limitations in terms of sensitivity and specificity for the diagnosis of CMV pneumonia. Although a high CMV viral load (VL) in plasma or whole blood is more likely to be associated with tissue-invasive CMV disease, CMV pneumonia can also occur with a low CMV VL in blood or whole blood plasma [22,23].

Consequently, direct measurement of CMV DNA in BAL samples has been considered a valid alternative in clinical practice for the diagnosis of CMV pneumonia. Some studies have correlated CMV VL in bronchoalveolar lavage fluid with CMV pneumonia but with inconsistent results [24,25].

The main contributor to this variability is the lack of a calibrator standard [26]; therefore, the World Health Organization (WHO) has developed an international standard for the calibration of HCMV QNAT, but, despite this, clinically relevant differences in values remain determined by the viral load from various HCMV QNATs [27].

As described, in our case, the BAL concentration was relatively low, with a value of 300 copies/mL and 2.4 log of CMV DNA, but it is known that several aspects can contribute to the variability of the viral load, such as the type of sample, the nucleic acid extraction method, the amplicon size, and the target gene [27,28].

Viral load values may also differ depending on the type of patient and their risk profile [29].

A 2017 study quantified CMV DNA in bronchoalveolar lavage fluid (BALF) of patients with CMV pneumonia who had previously undergone hematopoietic stem cell transplantation; they identified a cutoff level of 500 IU/mL to distinguish between pneumonia from CMV and lung shedding with a positive predictive value of 45%. However, different levels of VL may be appropriate in settings with very high or low prevalence of CMV pneumonia [30].

In fact, in another study on a small heterogeneous group of immunocompromised patients, using an FDA-approved PCR test on BALF, the difference between the CMV VL for the total group of immunocompromised patients and that for the group of transplant cases alone was 274 IU/mL compared to 34,800 IU/mL [6].

Histopathological examination on lung biopsy is considered the gold standard for the diagnosis of CMV pneumonia, due to the presence of nuclear inclusions that determine the characteristic owl’s-eye appearance of CMV in biopsy specimens [31].

However, the diagnostic result of lung biopsy may also vary as inclusions cannot always be visualized. CMV-specific immunohistochemical (IHC) staining in bronchial lavage fluid cytology specimens can detect CMV [32,33].

Ganciclovir and valganciclovir are the two antivirals successfully used in clinical practice for both the prevention and treatment of CMV infections.

The efficacy of immunoglobulin G for routine clinical use in the treatment of CMV pneumonia is unclear, although in patients undergoing hematopoietic cell transplantation, reductions in mortality appear to be primarily associated with advances in antiviral therapies and improvements in transplant practices, rather than immunoglobulin-based treatments [34].

Regarding the spread of CMV in patients with DS, an observational study demonstrated a high prevalence of CMV and high risk of CMV infection in children with DS who attended a nursery school for the mentally disabled [35].

Another recent retrospective study evaluated the incidence of all-cause pneumonia in subjects with DS, resulting in a higher probability of hospitalization and intensive care compared to those without DS. The mortality figure one year after the first pneumonia was also higher, with 5.7% versus 2.4% [8].

Pneumonia therefore disproportionately affects people with DS throughout their lives, starting from childhood, and represents an important cause of mortality. To date, however, there are no studies that have described the prevalence and severity of cytomegalovirus pneumonia in the population affected by DS.

The recent SARS-CoV-2 pandemic also confirmed a greater vulnerability of subjects with DS to develop more complications related to COVID-19, including ARDS. It has been shown that both children and adults with DS and SARS-CoV2 infection had longer hospital stays, longer periods of mechanical ventilation, a higher percentage of progression to sepsis, and increased mortality compared to patients without DS [36].

However, articles in the literature have only partially explained why the prevalence and severity of pneumonia and respiratory infections are greater in DS than in the general population, and the cause may be multifactorial.

People with DS have a greater susceptibility to infections, especially respiratory ones, due to anatomical anomalies of the airways, the presence of gastroesophageal reflux, and immune system disorders.

Immune disorders in DS remain poorly understood and vary from individual to individual. In some cases, they have been interpreted as a consequence of cellular oxidative damage responsible for premature aging, while in others, they have been considered as an intrinsic defect of the immune system [37].

Several genes involved in the function of the immune system are present on chromosome 21, but their overexpression has not been demonstrated [38]. Other genes linked to epigenetics could alter the expression and immune function of B or T cells [39].

The impaired response to vaccines with reduced levels of IgG2 and IgG4 also suggests a primary antibody deficiency. The proportions of B cell populations are similar to those found in common variable immunodeficiency, a condition that includes disorders characterized by hypogammaglobulinemia of unknown origin, inability to produce specific antibodies after immunization, and a susceptibility to bacterial infections, especially those by capsulated bacteria [38,40].

All available vaccinations are strongly recommended in subjects with DS to reduce the risk of infections and their complications, but only if vaccines generally induce a good antibody response [41]; some studies have demonstrated lower antibody titers and avidity compared to those for the population without DS due to intrinsic B cell defects, such as decreased total memory and immunoglobulin class switching, or with defects in both B cell and T cell responses [42]. Responses to the hepatitis B vaccine were weaker due to the greater presence of proinflammatory innate immune cells and higher levels of inflammation. It is therefore possible that the lesser longevity of vaccines against SARS-CoV-2 observed in individuals with DS could be linked to the increased baseline inflammatory levels described in these subjects. The administration of baricitinib, through inhibition of JAKq/JAK2, could reduce underlying inflammation in individuals with DS and restore immune homeostasis and the ability to trigger a more durable antibody response to vaccines [43].

Two vaccines for respiratory syncytial virus have recently been approved, and it may be useful to consider universal administration to all patients with DS. RSV infection in individuals with DS shows a higher prevalence and severity (measured by the need for hospitalization and length of hospitalization) compared to that in controls without DS [44]. Similarly, given the disease burden associated with pneumococcal pneumonia in patients with DS, there are gaps in knowledge of the causative agents, resulting in few studies on an adequate vaccination strategy and the efficacy of vaccination for pneumococcal pneumonia in children and adults with DS, as well as the costs [45,46].

Special attention is needed to increase adherence to timely vaccination recommendations for children with DS to optimize their health and potentially avoid hospital admissions. For example, the Parent Attitudes about Childhood Vaccines Survey (PACV) has proven to be a valuable tool for identifying vaccine hesitancy among parents of young children with DS [47].

There is currently no vaccine for CMV, although it is under development; therefore, personal hygiene measures are important, as is remaining vigilant about the potential etiology of CMV to be able to promptly diagnose and treat pneumonia in vulnerable subjects, such as subjects with DS. 

Future research may evaluate early identification through screening of urine and saliva specimens as for perinatal congenital cytomegalovirus (cCMV) infections. This could bring advantages in the management of patients with DS who are asymptomatic or with mild/moderate symptoms but who, as is known, can develop severe respiratory failure.

Recently, a study demonstrated that the Simplexa Congenital CMV Direct test can be used on saliva and urine samples from infants younger than 21 days to rule in or rule out congenital CMV infection quickly, sensitively, and specifically [48].

A positive saliva test confirmed by a urine test [49,50] allows institutions to easily use screening and confirmatory tests for cCMV. However, as with newborn screening, the combination of high false-positive rates and inconsistent shedding of CMV in saliva and/or urine could complicate the practice of CMV screening in vulnerable populations with high seroprevalence.

## 4. Conclusions

This case aims to highlight that CMV pneumonia in individuals with DS can be a life-threatening condition if not diagnosed and treated promptly.

The greater predisposition to the development of serious lung diseases in these patients could have a multifactorial etiology that includes the presence of immune disorders that are not yet well defined, especially in adulthood.

This case also highlights the importance and the limitations of prevention and rapid diagnostic testing for cytomegalovirus, as there are no well-defined viral load thresholds for populations with immune system disorders.

Further research on DS and respiratory infections as the main cause of death is therefore needed to provide more data on the prevalence of the etiological agents of pneumonia and to define better prevention strategies.

## Figures and Tables

**Figure 1 medicina-60-00242-f001:**
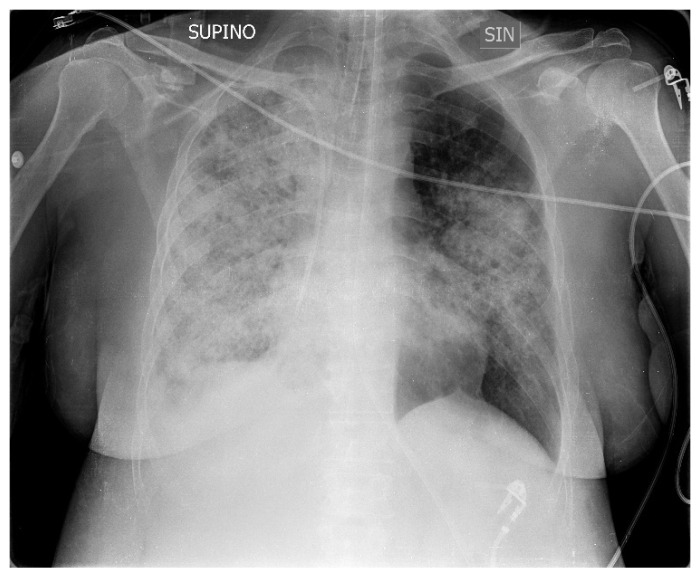
Chest X-ray upon admission to hospital shows bilateral pneumonia, with greater involvement of the right lung. SUPINO: supine; SIN: left.

**Figure 2 medicina-60-00242-f002:**
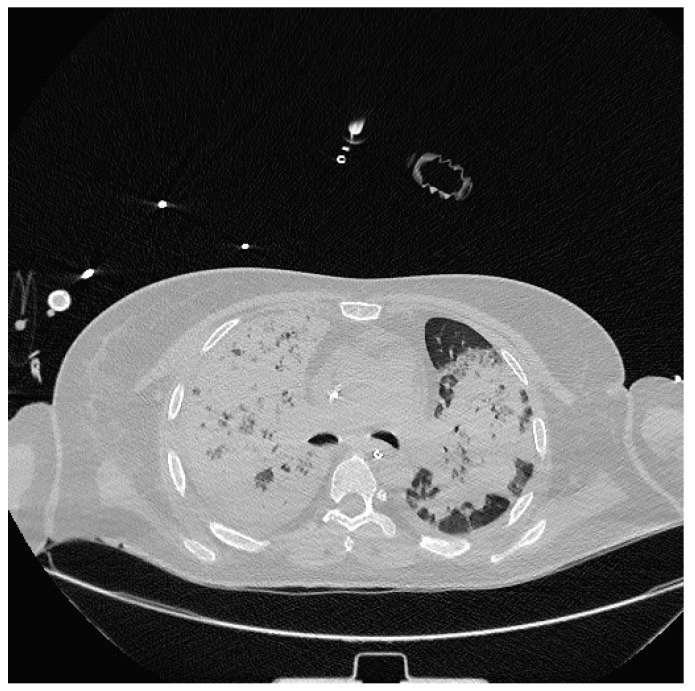
Chest CT showing extensive parenchymal thickening of the entire right lung and subtotal consolidation of the left lung.

**Figure 3 medicina-60-00242-f003:**
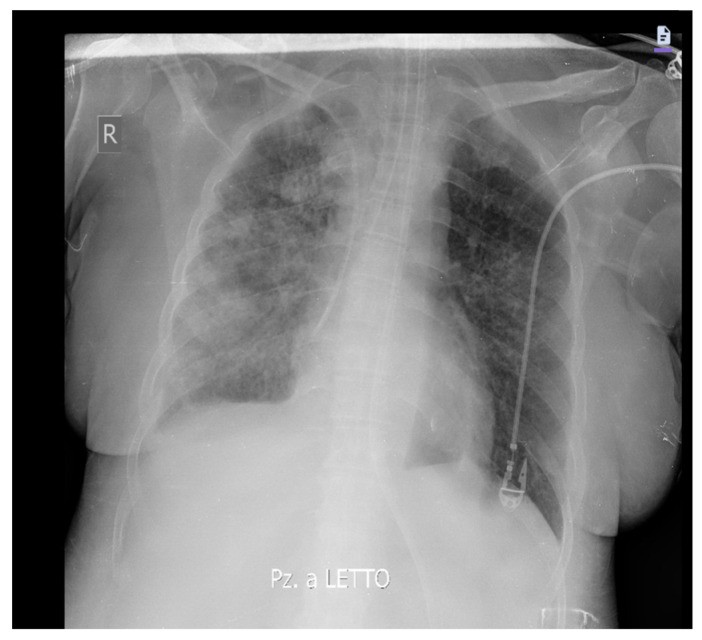
Chest X-ray 48 h after starting ganciclovir therapy shows an improvement in pulmonary densifications, more evident in the left lung.

**Figure 4 medicina-60-00242-f004:**
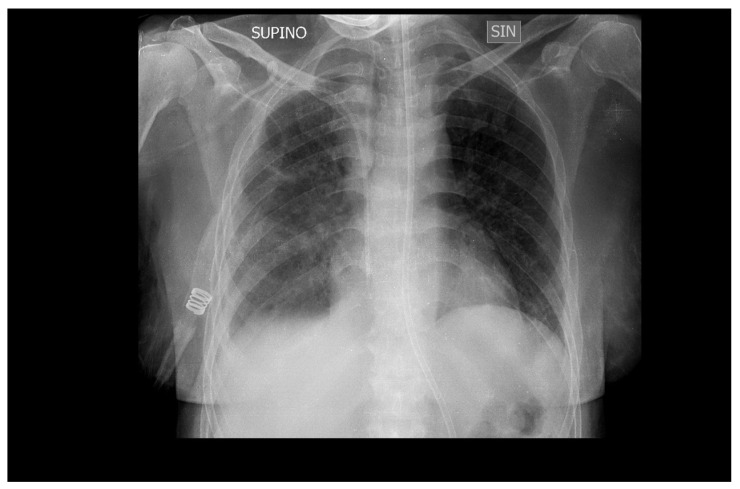
Chest X-ray of the patient extubated after 10 days of ganciclovir therapy shows almost complete resolution of bilateral interstitial pneumonia. SUPINO: supine; SIN: left.

**Table 1 medicina-60-00242-t001:** Initial increase in WBC and CRP values despite antibiotic and antifungal therapy and their subsequent progressive reduction after BAL and gancyclovir therapy.

DAY	1	2	3	4	5	6	7	8	9	10
WBC cells/^x^μL)	16,060	20,000	26,800	28,600	26,800	19,860	14,810	13,090	9750	7300
CRP (mg/L)	22.32	22.78	21.71	23.93	22.75	18.85	14.93	12.43	7.48	3.43
PCT (ng/mL)	3.22	3.17	2.65	1.87	1.32	1.0	0.83	0.89	0.74	0.7
CREA (mg/dL)	1.63	2.32	2.18	1.98	1.88	2.09	2.05	1.87	1.85	1.68
CMV DNA Copies/mL (BAL)				300 2.4 LOG						

WBC, white blood cells; CRP, C-reactive protein; CREA, creatinine; PCT, procalcitonin; BAL, bronchoalveolar lavage.

## Data Availability

Data are contained within the article.

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
