# Peer review of "Cytomegalovirus Pneumonia in a Patient with Down Syndrome"

_medicina, 2024, doi:10.3390/medicina60020242_

Round 1
Reviewer 1 Report
Comments and Suggestions for Authors
Thanks for coming up with a very nicely written case report.My few suggestions and clarifications are attached in manuscript, kindly address same.
Also,please mention whether the Consent from family obtained for publishing case report. Also any institutional approval obtained.

Minor mistakes in addressing as he to she etc.. which i think by accident.
Author Response
Thank you very much for taking the time to review this manuscript. I greatly appreciated your suggestions and proceeded to make the corrections you indicated. Changes are highlighted in yellow. Please see the attachment.
Reviewer 2 Report
Comments and Suggestions for Authors
Dear authors,
I have now completed the review of the manuscript titled "Cytomegalovirus pneumonia in Down Syndrome."
The case report is interesting and, in general, fair written.
The paper presents what it claims to be the first reported case of CMV pneumonia in a patient with Down Syndrome (DS), making it a potentially significant contribution to the medical literature. Also, given the increased susceptibility of individuals with DS to infections, especially respiratory ones, this case report adds valuable information on managing complex cases involving opportunistic infections like CMV.
Some suggestions are:
1. The report outlines the steps taken to diagnose CMV pneumonia, including the use of bronchoalveolar lavage and PCR testing for CMV-DNA. While the case details are thorough, it would be beneficial if the report discussed any challenges or considerations in the diagnostic process, especially given the rarity of the condition.
2. The report briefly discusses the prevalence of pneumonia in DS individuals and the general susceptibility of this population to respiratory infections. It would strengthen the report if it included a more extensive review of the literature, comparing this case to any similar instances or discussing how the management of this case aligns with or deviates from established protocols.
3. The paper emphasizes the importance of considering opportunistic pathogens like CMV in individuals with DS presenting with severe pneumonia. It highlights the need for early diagnosis and treatment. However, the report could further discuss long-term management strategies or preventive measures for such high-risk patients.
4. The report could provide suggestions for future research, such as studies on the prevalence of CMV pneumonia in DS or investigations into the best management strategies for similar cases.
In overall, the case report on CMV pneumonia in Down syndrome is a potentially valuable addition to the literature, especially for healthcare professionals encountering similar cases. It is well-detailed and provides a good basis for understanding this rare condition's clinical management. Future reports could benefit from a broader discussion of the literature, more comparative analysis with similar cases, and suggestions for future research directions to deepen understanding and improve care for this vulnerable population.
Thank you for your valuable contributions to our field of research. I will wait a revised manuscript.
Author Response
Thank you very much for taking the time to review this manuscript. I really appreciated your suggestions and I proceeded to discuss the points you indicated in more depth.I have discussed the diagnostic process in more depth. I considered a broader review of the literature, including similar cases such as SarsCoV2 infections in individuals with Down syndrome. I underlined the importance and "limitations" of vaccination and rapid diagnostic tests. Finally, I suggested the need to have data on prevalence and the possibility of carrying out screening with recent tests on saliva and urine.
Corrections made are underlined in green Please see the attachment
